# “What Do You Get? Nothing”: A Qualitative Analysis of the Financial Impact of Family Caregiving for a Dying Relative at Home in Germany

**DOI:** 10.3390/healthcare13070810

**Published:** 2025-04-03

**Authors:** Sally Pieper, Alina Kasdorf, Raymond Voltz, Julia Strupp

**Affiliations:** 1Department of Palliative Medicine, Faculty of Medicine and University Hospital, University of Cologne, Kerpener Str. 62, 50937 Cologne, Germany; alina.kasdorf@uk-koeln.de (A.K.); raymond.voltz@uk-koeln.de (R.V.); julia.strupp@uk-koeln.de (J.S.); 2Center for Integrated Oncology Aachen Bonn Cologne Duesseldorf (CIO ABCD), Faculty of Medicine and University Hospital, University of Cologne, Eupener Str. 129, 50933 Cologne, Germany; 3Center for Health Services Research, Faculty of Medicine and University Hospital, University of Cologne, Kerpener Str. 62, 50937 Cologne, Germany

**Keywords:** dying at home, family caregivers, financial loss, caregiver burden, end of life

## Abstract

**Background/Objectives**: As a result of demographic change in Germany, the number of people in need of care is steadily increasing, with a correspondingly larger proportion of care being provided by family members at home. Family caregivers face significant challenges in providing such care, particularly when balancing work responsibilities. Many experience a loss of income due to reduced working hours or the necessity of leaving the labor market. Additional caregiving costs, such as medical expenses, transportation, and home modifications, further exacerbate their financial burden. **Methods**: This study consists of an online survey, which included the German version of the Carer Support Needs Assessment Tool (CSNAT), designed to assess the support needs of family caregivers. Respondents were asked to describe their support needs in open-text responses. To illustrate their experiences, a qualitative content analysis was conducted. **Results**: Out of the 320 questionnaires, 304 of them contained open-text responses that could be analyzed. Important themes included the need for support in the workplace, financial security, and assistance with administrative barriers. In addition to more flexible working hours and greater understanding from employers, the need for financial compensation for loss of working time was expressed. **Conclusions**: Despite a growing awareness of the gaps in support, the needs of family caregivers remain inadequately addressed, leaving them financially burdened and unsupported. Ultimately, this study calls for a re-evaluation of societal attitudes toward caring, arguing for greater recognition of the economic contributions of family caregivers and the implementation of supportive policies.

## 1. Introduction

The global health care landscape is facing significant challenges due to demographic shifts, particularly aging populations and increasing chronic diseases [1,2]. This trend is straining health care systems worldwide, necessitating structural adaptations and a shift towards home-based care [3].

The number of people in need of care in Germany has risen sharply in recent years due to demographic change. In December 2023, 5.69 million people needed care, of whom around 86% (4.49 million) were cared for at home—mainly by relatives (67%) [4]. Indeed, care provided by family caregivers plays a central role in the German health care system: It is estimated that around 5 million people in Germany provide care to relatives at home [5], with women more likely to provide care [6]. Overall, between 5% and 6% of adults provide care on a regular basis [7].

One of the main contributors to home care are the terminally ill, as many people prefer to spend the last days and hours of their lives at home and to die there [8], which means that end-of-life care tasks are shifting from medical care institutions to family members. This development poses considerable challenges for family caregivers [9,10]. In particular, reconciling care and work is becoming increasingly problematic [11], as the proportion of working family caregivers under the age of 65 has risen from 53% to 66% [7]. This is particularly significant when accounting for the length of care, as 64% of care situations last longer than a year [12]. The baby boomer generation’s retirement will exacerbate this situation, leaving fewer working age individuals to support a growing number of care-dependent people [13,14]. This combination of care recipients ‘desire to be cared for and die at home and family caregivers’ work commitments creates additional burdens that require extensive support [15].

The financing of home-based end-of-life care in Germany relies primarily on statutory long-term care insurance, supplemented by health insurance and out-of-pocket expenses. Depending on their level of dependency, people in need of long-term care receive cash allowances (“Pflegegeld”) for caregiving or in-kind benefits (“Pflegesachleistungen”) for professional home care [16,17]. However, gaps in coverage persist, often leaving families with additional costs for home adaptations or extended care [18]. Fewer than 10% of people in need of home care do not have family caregivers involved in their care [19]. This means that family caregivers provide the majority of all care services today, making them a crucial social and economic resource for the health care system [19,20,21]. Among those who provide familial care, financial difficulties are widespread, with 38% of family caregivers reporting moderate to high financial stress and 45% experiencing at least one financial impact, such as reduced savings or increased debt [22]. In addition, recent studies have shown that family caregivers are often forced to reduce their working hours or take short-term sick leaves [23].

In Germany, there is limited research from the perspective of family members regarding their burdens, especially concerning their financial strains. Therefore, our study aimed to investigate the financial impact of caring for a seriously ill and dying family member in Germany by using a panel. Our study is part of a larger project called “Dying at Home”. The results of the interviews and a large online survey have been published elsewhere [23,24].

## 2. Materials and Methods

### 2.1. Study Design, Population, and Data Collection

The online observational panel survey was designed to be completed via a computer or mobile device with open-text options.

The survey consisted of three parts: (i) items generated from a qualitative multi-method study combining the results of semi-structured interviews and focus group discussions to explore the support needs of family caregivers of patients wishing to die at home; (ii) a German version of the Carer Support Needs Assessment Tool (CSNAT) (called KOMMA in German (The KOMMA assessment form (communication with relatives) was used in this study (Copyright 2019 University of Cambridge/The University of Manchester, all rights reserved). The KOMMA was translated by Sabine Pleschberger, Johannes Bükki, and Christiane Kreyer from the original version, and the CSNAT was developed by Dr. Gail Ewing and Prof. Gunn Grande (Copyright @ 2009 University of Cambridge/The University of Manchester, all rights reserved)) [25,26], and (iii) socio-demographic and patient characteristics. This publication focuses on the exploratory analysis of open-text comments, while the remaining data and results of the survey, including a detailed description of the survey methods, have been analyzed and published elsewhere [23].

This study was prospectively registered in the German Clinical Trials Register (DRKS00026229) on 25 November 2021. Ethical approval was granted by the Ethics Committee of the Medical Faculty of Cologne (#21-1466). Family caregivers were included in this study if they were older than 18 years and had given written consent to participate in this study. Deaths before 2016 were excluded to minimize recall bias. The survey was open from 19 September 2022 to 13 November 2022.

The survey was conducted anonymously, with the first section providing detailed information on data protection and obtaining informed consent from participants. After consent was given, no data sensitive to privacy were collected. Participants were assigned an ID, ensuring that the dataset contained no names or personally identifiable information.

The 14 questions of the CSNAT, which were used retrospectively, asked about the family members’ need for support in different areas of life, such as knowledge about the disease, time for themselves, resources, legal and professional issues, caregiving, and day and night help. Participants in the online survey were able to choose between the response options “No”, “A little more”, “Much more”, and “Significantly more”. The research team (AK, JS) added an open-text option to each of the items. The optional open-text provided an opportunity to specify the type of support the person would have liked, allowing for a more precise consideration and, if necessary, the addition of individual needs. This procedure was agreed upon with the KOMMA team prior to the start of the survey.

### 2.2. Analysis

As part of the qualitative analysis, all open-text comments on support needs collected in the questionnaires were entered into MAXQDA 2024 in order to carry out systematic coding. Using content analysis, we analyzed the open-text content and developed a category system with main and subcategories in a multi-step process [27,28]. The open-texts were coded and analyzed by SP in consultation with AK. S.P. and A.K. reviewed and discussed the codes and categories to reach consensus, making adjustments as needed. First, relevant text passages were identified based on their connection to the research question. By analyzing these passages, preliminary codes were formed inductively. It became clear that some participants expressed a similar need for additional support in several areas.

As part of the inductive coding process, item 31, which deals with professional, legal, and financial support, was examined in more detail. After coding all text passages, they were sorted thematically. Main and subcategories were created for a higher level of abstraction. SP and AK repeatedly consulted each other to discuss the codes critically. The analysis made it possible to develop specific codes within each main category in order to capture more precisely the different ways in which support needs were expressed. The team selected exemplary open-text quotations based on representativeness and expressiveness of their content.

## 3. Results

### 3.1. Survey Sample

From a total of 320 questionnaires collected in this study, the open-text data of the informants could be analyzed in 304 cases, which means that 95% of the informants added comments to at least one of the available open-text fields in their questionnaires [23]. Women made up the majority of the sample, representing 70.7% of participants (Table 1).

### 3.2. Open-Text Analysis

A total of 133 respondents expressed a great need for support in the areas of financial, legal, and professional matters. Of these, 40% stated that they needed much more support, and 21% needed significantly more support. The open-text comments were organized into three major categories with subcategories (Table 2).

#### 3.2.1. Employer Support

Participants expressed specific wishes and needs for their employers in terms of support that would enable them to better reconcile their work commitments with their caregiving responsibilities. This category was divided into the following sub-categories:Employer flexibility: One of the participants’ requests was to be able to organize their working hours more flexibly in order to make it easier to combine work and caring responsibilities. Reflecting this wish, one participant noted, “*Less working hours, more flexible working hours*” (Respondent-ID: 254).Sick leave/leave of absence for care: In order to fulfill caring responsibilities at home, participants emphasized that they sometimes have to take sick leave. One of the respondents commented on this: “*For example, I took sick leave to be there for my grandmother. I’m sure there are other ways*” (Respondent-ID: 210). This quote emphasizes the lack of options for time off, forcing participants to rely on other options, such as sick leave.Understanding of the employer: This sub-category highlighted the need for more understanding from participants’ employers. Specifically, respondents often described employers showing little consideration for their caring responsibilities: “*If my employer had also accepted family caregiving leave. Unfortunately, I work in a care home and was called in during my free time*” (Respondent-ID: 108).Financial compensation due to loss of work: A key issue was the financial loss suffered by participants who worked while caring for a seriously ill relative. One family carer said, “*Unable to work because of caring (24-h on-call duty)! Financial support for family caregivers too!!!*” (Respondent-ID: 42). Another participant wrote, “*Financial compensation for family caregivers who give up years of their lives and are unable to go to work*” (Respondent-ID: 57).

#### 3.2.2. Financial Support

Two sub-categories were identified in this area, focusing on the financial security of participants:More care allowance: With regard to the carer’s allowance, there was a need for comprehensive cost coverage by the care insurance funds. One respondent said, “*That the care insurance fund covers everything you need*” (Respondent-ID: 175).Financial security: A key aspect in the area of financial support is financial security. The following quote from a caring relative underlines that financial pressure also had an impact on their own health: “*What do you get? NOTHING. Caring relatives should be financially and socially secure so that they have a clear head for caring. Maybe then fewer of them will fall ill themselves, because they are carrying a double and triple burden*” (Respondent-ID: 63). The challenge of organizing professional care without sufficient financial resources was also discussed: “*but we were too poor to pay for night care or personal care through a care service*” (Respondent-ID: 88).

#### 3.2.3. Administrative Barriers

This area represented two overarching aspects that were reflected in both the professional and financial contexts. Many participants reported a high level of bureaucracy and a lack of adequate advice, which was an additional burden.

Information and advice: A key aspect of this category was the lack of knowledge and advice, particularly in relation to the types of support available, whether financial or care-related. Participants emphasized the importance of being informed about possible support: “*Providing knowledge*, *help with applications, etc. What is possible at all—financial support*, *care support*, etc.”. (Respondent-ID: 256).Bureaucratic barriers: The need for better support in overcoming bureaucratic barriers was repeatedly emphasized by family caregivers. In particular, lengthy and often frustrating dealings with authorities were described as a significant burden: “*Battles with authorities*, *social applications*, *many hours invested that you don’t really have*” (Respondent-ID: 184).Difficulties with care levels: Dissatisfaction with the classification into care levels was another issue raised by participants. They reported discrepancies between their relative’s actual care needs and the care categorization they were given, which they felt did not reflect their individual needs: “*The carer who did the categorization for care was probably blind and deaf. Or just completely idiotic. My father couldn’t walk five steps without gasping for breath, he couldn’t wash himself or prepare his medication, let alone take it. He was totally dependent on help. And what did he get? With a lot of hanging and asking, just level 1. That kind of thing is a cheek [meaning ridiculous]*” (In Germany, the previous levels of care were replaced by five levels of care as of 1 January 2017. This categorization reflects a person’s level of independence and determines the corresponding care benefits. The categorization is carried out by the care insurance company on the basis of an expert opinion that determines the individual’s need for support [29]) (Respondent-ID: 119).Assistance with documents: Dealing with the amount of paperwork was a significant challenge for family caregivers. Filling in forms and applications was perceived as very stressful. Many respondents said that more support was needed in this area to reduce the bureaucratic burden: “*Support in dealing with the mountain of paperwork, applications, etc.*” (Respondent-ID: 58).

These subcategories helped us better recognize specific differences within the overarching area of “financial, legal, or professional matters” and illustrate the complexity of needs.

## 4. Discussion

This study aimed to explore the financial impact of caring for a seriously ill and dying family member at home. The findings highlight the significant challenges faced by family caregivers and the need for comprehensive support, particularly in the areas of workplace policies and financial security.

One noteworthy aspect of this care dynamic concerns gender disparities, as our findings are in line with wider research on the persistence of gender inequalities in the assumption of care work [30]. Women of working age are more likely to provide care and spend more time providing care than men, often while maintaining their employment [31]. This “gender care gap” leads to restrictions on women’s life choices and earning potential [30].

The age of panel participants in this study appears to be younger than in other studies. This difference suggests that the unique focus on balancing work and caring for a dying relative may have attracted family caregivers with a demographic less represented in traditional self-selected samples, highlighting a critical but often overlooked perspective on intergenerational care dynamics.

Our findings show that family caregivers face several critical barriers, including balancing work commitments, financial insecurity, and bureaucratic barriers. These factors not only impede their ability to provide care but also affect their overall well-being [32]. It is essential to recognize and address these issues through appropriate resources and support systems [33,34].

The health care reform in Denmark in 2007 shows that systematic changes are possible. Focusing on specialization, centralization, and digitalization, the reform has attracted international attention [35]. It significantly restructured the health sector, with the specific aim of improving the transparency and quality of home care [36,37]. The Danish system stands out among European countries for its comprehensive coverage: 59% of those in need of care receive state-provided care [38]. As a result, Denmark is among the countries with the lowest rates of daily care, similar to other Nordic countries, where individuals requiring such care benefit from the well-developed formal long-term care sector and better paid coverage [39]. While some countries, particularly in Northern Europe, have developed robust policies and services for caregivers [40], others, particularly in the Global South, lack comprehensive support systems [41]. Cross-country studies highlight differences in caregiver experiences, with those in countries offering more developed formal support generally reporting lower burden and higher well-being [42]. Workplace policies to support caregiving vary internationally, with some countries implementing innovative approaches such as shared paid leave systems [43].

In particular, family caregivers of terminally ill patients often face significant out-of-pocket expenses, including medical supplies, medications, and transportation costs for medical visits. According to a recent study by Hastert et al. [44], logistical costs were the most common out-of-pocket expenses (58%), followed by medication costs (35%) and medical bills (17%). Furthermore, severe financial hardship was reported by 38% of family caregivers. Such financial burdens can lead to increased debt and reduced savings, resulting in long-term economic hardship.

Family caregivers are an important social and economic resource for the health care system [19,20,21], and caring should not negatively affect the economic future of family caregivers. Many family caregivers who work are forced to reduce their working hours or interrupt their careers to provide care, with both short- and long-term financial consequences [45]. A recent study by Kasdorf et al. [23] found that 28% of respondents had to take sick leave to care for a relative. In addition, 40.4% reported having to use their savings to pay for care, and 36.9% reported that their net monthly income was insufficient to cover the cost of care [23]. Long-term effects may include difficulties returning to work or loss of pension benefits. To mitigate these effects, frameworks need to be implemented that allow family caregivers to maintain both their caring responsibilities and their professional and financial futures [45]. These caregivers should be able to combine work and care by benefitting from financial support, flexible working hours, and the opportunity to work from home [46]. Instead, the lack of support in the workplace emerged as a key concern for many family caregivers. Participants expressed a strong desire for more adaptable working hours to balance caring and work responsibilities, a finding that is consistent with the wider research [47,48]. In addition, the need for leave options and greater employer understanding was highlighted, underlining a gap in employer support. Policies such as carer’s leave could help family caregivers take the time they need without fear of losing their jobs or income. In order to facilitate a sensitive approach to serious illness, death, and bereavement and to promote communication and emotional support among colleagues at work, training programs in bereavement support, such as those offered by the LAUT project in Germany (“Letzthelfer:innen am Arbeitsplatz-für einen sensiblen Umgang mit Sterben, Tod und Trauer (LAUT),” available at https://letzthelferamarbeitsplatz.com/ (accessed 3 February 2025)), should be implemented [49].

Additionally, participants in our study expressed a strong need for a more comprehensive financial safety net, as many face significant economic hardship due to reduced working hours or an inability to work. Financial stress, in turn, can have a negative impact on the physical and mental health of family caregivers [22,50]. In addition, many participants reported that they could not afford professional care services, highlighting the financial barriers to accessing adequate support [51].

Bureaucratic barriers and a lack of information about available financial resources compound the stress family caregivers face [52]. The time-consuming and often confusing application processes for financial and care-related support contribute to family carer burnout and feelings of helplessness [53]. Research suggests that proactive support is essential to meet the needs of family caregivers. A “buddy” system, where knowledgeable individuals link families to health and social care services, could bridge the gap between family caregivers’ needs and available support [54]. Raising awareness of family caregiving in both social and professional settings is also essential, especially as demographic changes increase the need for care [21]. Policy makers should prioritize the inclusion of these recommendations in their policy considerations [32].

In sum, our findings highlight several areas where improvements are urgently needed. Changes in workplace policies, financial support systems, and bureaucratic processes are required to better support family caregivers. Policies that promote flexibility in the workplace, ensure adequate financial compensation, and reduce bureaucratic barriers are crucial to reducing the burden on family caregivers. In addition, clearer information and guidance on available services and benefits will help family caregivers navigate the complexities of caregiving and ultimately improve the quality of care for their loved ones.

### Strengths and Limitations

To our knowledge, this is the first study in which the German version of the CSNAT has been used retrospectively in a questionnaire and expanded to include open-text comments, which enables a more precise analysis of family caregivers’ needs. The large sample size permitted a meaningful assessment of family caregivers’ needs in addition to the quantitative data collection, which has been published elsewhere [23]. The qualitative analysis provided a detailed analysis of individual statements, giving family caregivers a voice. The use of a panel achieved a high response rate, ensuring the representativeness of the results across different conditions and regions of Germany. There is, though, a potential risk of self-selection, as 65% of survey participants were panel members. Furthermore, as the survey was conducted online, participants’ digital competences may have influenced the results differently depending on their age, with older participants potentially having less experience with online surveys. This may partly explain the differences in the age distribution of our sample compared to other studies. Although our retrospective research design, which asked bereaved family caregivers to report on their experiences of end-of-life care, may have limitations, such as possible recall bias and the influence of emotion, studies have shown that brief events are often accurately recalled [8].

The validity of the CSNAT for family caregivers in palliative care has been demonstrated internationally [55]. The domains of the CSNAT comprehensively cover the different areas of support needs, and it serves as a tool in research and everyday palliative care [25].

## 5. Conclusions

Although the gaps in support for family caregivers have long been recognized, the individual needs of family caregivers and the needs-based implementation of effective interventions have not been sufficiently explored. The burden on family caregivers remains high, and families carry much of the burden with seemingly no financial support. As home care becomes more prevalent, financial support for family caregivers is essential to meet the increasing demands of end-of-life care at home. Key measures include the introduction of income replacement benefits, similar to parental leave, to mitigate income loss during caregiving periods. In addition, the recognition of caregiving time for pension purposes can help prevent long-term financial disadvantages. A more flexible and less bureaucratic use of respite care services, such as short-term care and relief care, is also crucial to provide family caregivers with the breaks they need. Widening access to these services and increasing the availability of community-based and digital support, complemented by better advice on existing support services through practical helpers, such as a “Buddy in the last year of life” or “Last-Aiders at work”, can further improve family caregivers’ access to practical, emotional, social, and financial support.

## Figures and Tables

**Table 1 healthcare-13-00810-t001:** Demographics and characteristics of study population (subgroup n = 304/320).

Characteristics	Total	
	n	%
Family caregivers		
Age		
Mean (SD, Min—Max)	50.5 (13.7, 18–82)	
Gender		
Male	86	28.3
Female	215	70.7
Diverse	3	1.0
Relationship		
Spouse/Partner	33	10.9
Son/Daughter	125	41.1
Brother/Sister	9	3.0
Son/Daughter-in-law	20	6.6
Grandson/-daughter	47	15.5
Friend	18	5.9
Neighbor	20	6.6
Volunteer	11	3.6
Other	21	7.0
Employment status *		
Was employed in the last 3 months	197	67.7
Was not employed in the last 3 months	94	32.3
Deceased		
Age		
Mean (SD, Min–Max)	77.5 (13.8, 19–105)	
Gender		
Male	156	51.3
Female	147	48.4
Diverse	1	0.3
Diagnosis **		
Cancer	111	36.5
Neurological disease	95	31.3
Cardiovascular disease	104	34.2
Respiratory disease	42	13.8

* Total n = 291, missing or not applicable n = 13 ** Multiple response possible.

**Table 2 healthcare-13-00810-t002:** Overview of categories and subcategories.

Category	Subcategory
Employer support	Employer flexibility
	Sick leave/leave of absence for care
	Understanding of the employer
	Financial compensation due to loss of work
Financial support	More care allowance
	Financial security
Administrative barriers	Information and advice
	Bureaucratic barriers
	Difficulties with care levels
	Assistance with documents

## Data Availability

Selected parts from the original contributions presented in this study are included in the article. Further inquiries can be addressed to the corresponding author.

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
