# Peer review of "“What Do You Get? Nothing”: A Qualitative Analysis of the Financial Impact of Family Caregiving for a Dying Relative at Home in Germany"

_healthcare, 2025, doi:10.3390/healthcare13070810_

Round 1
Reviewer 1 Report
Comments and Suggestions for Authors
The article addresses a very relevant topic related to population ageing and "welfare" responses that come from inside the family.
As of great interest, the article deserves publication as long as these revisions will be made:
- The title must be shortened to be more captivating. I would suggest avoiding using two sentences with two colons. It's too articulated;
- In methodology section Authors should specify where the online survey was available;
- Authors should pay attention to the use of punctuation. In paraghraphs 3.2.1, 3.2.2 and 3.2.3. be aware thet after ":" no capital letter is ever needed. Please switch to lowercase letters and separate each sentence with semicolons ";" and not "." .
- line 123 please remove ".
Finally Authors should briefly clarify the anonymization process in the responses analyses.
After these modifications being made I would reccomend publication.
Comments on the Quality of English LanguageGrammar revision needed.
Author Response
Thank you very much for taking the time to review this manuscript. Please find the detailed responses below and the corresponding revisions/corrections highlighted/in track changes in the re-submitted files.
Comment 1: The title must be shortened to be more captivating. I would suggest avoiding using two sentences with two colons. It's too articulated.
Response 1: Thank you for your comment. The title has been shortened accordingly to make it more captivating.
Comment 2: In methodology section Authors should specify where the online survey was available.
Response 2: Thank you for your comment. To avoid self-plagiarism, the detailed description of the online survey has not been repeated and can be found in Kasdorf et al. 2024 (Dying at home: What is needed? Findings from a nationwide retrospective cross-sectional online survey of bereaved people in Germany)
Comment 3: Authors should pay attention to the use of punctuation. In paraghraphs 3.2.1, 3.2.2 and 3.2.3. be aware thet after ":" no capital letter is ever needed. Please switch to lowercase letters and separate each sentence with semicolons ";" and not ".".
Response 3: Thank you for your comments. The text has been revised accordingly.
Comment 4: line 123 please remove ".
Response 4: Thank you for your comment. The unnecessary punctuation has been removed.
Comment 5: Finally Authors should briefly clarify the anonymization process in the responses analyses.
Response 5: Thank you for your suggestion. We have added more information to this section.
Comment 6: Comments on the Quality of English Language: Grammar revision needed.
Response 6: The manuscript has been professionally proofread, the certificate is attached for reference.

Reviewer 2 Report
Comments and Suggestions for Authors
Authors: Thank you for the opportunity to review this manuscript. This was an excellent manuscript and the topic is of critical importance, globally. I personally enjoyed getting to understand the caregiving landscape in another country. Please see below the view comments/edits I have suggested for this manuscript.
Introduction
Line 64-66: I would consider making this sentence two separate sentences. It is a lengthy sentence and at first read, I was not clear what the purpose of this sentence was.
Methods
Analysis
Line 115: You can remove the word “qualitative” before content analysis
Line 116: You can remove the word “qualitative” before category system
Results
Survey sample
Line 130-131: I would re-word this sentence. The phrasing of it makes it seems as if you are going to talk about 70% of the sample, when what you mean is that 70% of the sample are women.
Author Response
Thank you very much for taking the time to review this manuscript. Please find the detailed responses below and the corresponding revisions/corrections highlighted/in track changes in the re-submitted files.
Comment 1: Introduction
Line 64-66: I would consider making this sentence two separate sentences. It is a lengthy sentence and at first read, I was not clear what the purpose of this sentence was.
Response 1: Thank you for your comment. The text has been revised accordingly.
Comment 2: Methods
Analysis
Line 115: You can remove the word “qualitative” before content analysis
Line 116: You can remove the word “qualitative” before category system
Response 2: Thank you for the suggestion, we changed it accordingly
Comment 3: Results
Survey sample
Line 130-131: I would re-word this sentence. The phrasing of it makes it seems as if you are going to talk about 70% of the sample, when what you mean is that 70% of the sample are women.
Response 3: Thank you for the suggestion, the sentence has been adjusted.
Reviewer 3 Report
Comments and Suggestions for Authors
Dear Authors,
thank you for the opportunity to review this important contribution to the topic. Here, are some points to improve the scientific soundness of your manuscript:
1) The title is clear and informative but could be more concise. While it effectively communicates the scope of the study, it may benefit from simplification. For example, "Exploring the Financial Impact of Family Caregiving for a Dying Relative at Home in Germany: A Qualitative Analysis" or "The Financial Burden of Family Caregivers for Terminally Ill Relatives at Home: A Qualitative Study Using the Carer Support Needs Assessment Tool in Germany" would still capture the main idea while making the title more streamlined.
2) There are indeed references that are older than five years (2014-2020). More recent literature on caregiver support should be included, especially given the evolving nature of these topics. It would be beneficial for the authors to include more recent studies (last 5 years) that discuss caregiver support or similar financial impact studies, as these topics evolve with changing societal and policy contexts.
3) Some parts of the introduction are lengthy and contain excessive background details. This could be condensed to ensure the reader's attention is focused on the main objectives of the study. Please, provide a more concise overview of the demographics and the state of family caregiving in Germany to make the research gap more apparent. It is also recommended (at the beginning of your Introduction) to draw a parallel with an international perspective on the same issue, particularly regarding demographic and epidemiological transitions. A recent editorial (PMID: 39169799) provides valuable insights into how these transitions—such as an aging population and rising healthcare demands—are shaping patient needs and impacting healthcare systems and caregiving. This reference will strengthen your discussion by emphasizing the increasing complexity of global healthcare needs and the urgent need for healthcare systems to adapt to these changes.
4) While the use of the Carer Support Needs Assessment Tool (CSNAT) is well explained, the paper lacks a bit of detail on the online survey’s implementation and how potential biases were mitigated, especially since the study relies on self-reported data. Moreover, the data analysis section mentions MAXQDA 2024, but there is limited description of how the data were coded, which would strengthen the methodology section. Please, include a clearer description of the coding process in the methodology section, explaining the categorization in more detail.
5) Discussion section: there is some overlap in the themes discussed, particularly around financial stress and work-life balance. These sections could be condensed to avoid repetition. Additionally, please integrate more international perspectives on caregiver support systems and compare the results with studies from countries with more robust caregiver support policies.
Author Response
Thank you very much for taking the time to review this manuscript. Please find the detailed responses below and the corresponding revisions/corrections highlighted/in track changes in the re-submitted files.
Comment 1: The title is clear and informative but could be more concise. While it effectively communicates the scope of the study, it may benefit from simplification. For example, "Exploring the Financial Impact of Family Caregiving for a Dying Relative at Home in Germany: A Qualitative Analysis" or "The Financial Burden of Family Caregivers for Terminally Ill Relatives at Home: A Qualitative Study Using the Carer Support Needs Assessment Tool in Germany" would still capture the main idea while making the title more streamlined.
Response 1: Thank you for your valuable feedback and suggestions. The title has been shortened to make it more concise and streamlined.
Comment 2: There are indeed references that are older than five years (2014-2020). More recent literature on caregiver support should be included, especially given the evolving nature of these topics. It would be beneficial for the authors to include more recent studies (last 5 years) that discuss caregiver support or similar financial impact studies, as these topics evolve with changing societal and policy contexts.
Response 2: Thank you for your comment. We have included more recent literature. Please see tracked changes within the manuscript.
Comment 3: Some parts of the introduction are lengthy and contain excessive background details. This could be condensed to ensure the reader's attention is focused on the main objectives of the study. Please, provide a more concise overview of the demographics and the state of family caregiving in Germany to make the research gap more apparent. It is also recommended (at the beginning of your Introduction) to draw a parallel with an international perspective on the same issue, particularly regarding demographic and epidemiological transitions. A recent editorial (PMID: 39169799) provides valuable insights into how these transitions—such as an aging population and rising healthcare demands—are shaping patient needs and impacting healthcare systems and caregiving. This reference will strengthen your discussion by emphasizing the increasing complexity of global healthcare needs and the urgent need for healthcare systems to adapt to these changes.
Response 3: Thank you for your comments. We have shortened the introduction accordingly and provided more details on demographics and the state of family caregiving in Germany. We also appreciate your suggestion regarding the editorial. Although the development of clinical nursing information systems is highly relevant and becoming increasingly important due to demographic changes, we believe that it may not offer significant benefits for our paper at this stage.
Comment 4: While the use of the Carer Support Needs Assessment Tool (CSNAT) is well explained, the paper lacks a bit of detail on the online survey’s implementation and how potential biases were mitigated, especially since the study relies on self-reported data. Moreover, the data analysis section mentions MAXQDA 2024, but there is limited description of how the data were coded, which would strengthen the methodology section. Please, include a clearer description of the coding process in the methodology section, explaining the categorization in more detail.
Response 4: Thank you for your valuable feedback. We acknowledge the importance of providing more details on the online survey’s implementation and our approach to mitigating potential biases. To ensure the feasibility and objectivity of the survey, we conducted pre-tests prior to the full implementation. Additionally, potential recall bias due to the retrospective nature of the survey was considered. However, previous studies indicate that recall bias in similar contexts is minimal (Dust et al. 2022). To further reduce recruitment bias, a stratified sampling approach was employed based on a predefined categorization of target groups within the overall population. In order to avoid self-plagiarism, the detailed description of the implementation of the online survey was not repeated and can be found in Kasdorf et al. 2024 (Dying at home: What does it need? Findings from a nationwide retrospective online cross-sectional survey of bereaved people in Germany) The survey itself was designed to be completed on both computers and mobile devices and consisted of three parts: (i) items derived from a prior qualitative multi-method study, including semi-structured interviews and focus groups, which explored the support needs of primary caregivers of patients wishing to die at home (for details, see Kasdorf et al. 2023), (ii) a validated German version of the Carer Support Needs Assessment Tool (CSNAT) (Ewing et al. 2013; Kreyer et al. 2020), and (iii) sociodemographic and patient-related characteristics. Furthermore, we have revised the data analysis section to provide a more detailed description of the coding process in MAXQDA 2024. Specifically, we now outline our structured qualitative content analysis approach, including the inductive development of categories and reliability checks. These revisions enhance the transparency and methodological rigor of our study.
Comment 5: Discussion section: there is some overlap in the themes discussed, particularly around financial stress and work-life balance. These sections could be condensed to avoid repetition. Additionally, please integrate more international perspectives on caregiver support systems and compare the results with studies from countries with more robust caregiver support policies.
Response 5: Thank you for the comment. The repetition regarding financial stress and work-life balance has been removed. More international perspectives on caregiver support systems have been integrated into the text. Please see tracked changes within the manuscript.
Reviewer 4 Report
Comments and Suggestions for Authors
The manuscript aimed to investigate the financial impact of caring for a seriously ill and dying family member in Germany by using a panel, a crucial topic which is impacting many societies. The results come from a larger study, conducted through an online observational panel survey, focused on assessing perspectives of family members regarding their burdens, especially concerning their financial strains. The manuscript focuses on qualitative analysis of the open-text comments on support needs provided by family caregivers.
Here some comments that could potentially improve the already good paper:
- Authors stated that “The use of a panel achieved a high response rate, ensuring the representativeness 300 of the results across different conditions and regions of Germany” (p.8 lines 300-301). How did the researchers generate the sample? Which is the population target (nation wide or regional)? Does the sample statistically represent the population studied?
- Could authors provide more information about the services provided to the deceased (and caregivers) during the care pathways? This comment is due to the different mix of aides provided by different services: for instance, home palliative care services could differ from geriatric home care services, especially regarding bureaucratic/administrative barriers, composition of the care teams and 24/7 assistance.
- Could authors provide any information about the dimensions of the trade-off regarding the resources provided to caregiver/potentially needed from the caregiver-patients?
- I would suggest that authors should emphasize, in Strength and Limitations section, that since the survey has been conducted online, alphabetization on using online survey could affect differently according to the age. This aspect may explain partially the differences in age distribution of the sample, compared to other studies.
Author Response
Thank you very much for taking the time to review this manuscript. Please find the detailed responses below and the corresponding revisions/corrections highlighted/in track changes in the re-submitted files.
Comment 1: Authors stated that “The use of a panel achieved a high response rate, ensuring the representativeness 300 of the results across different conditions and regions of Germany” (p.8 lines 300-301). How did the researchers generate the sample? Which is the population target (nation wide or regional)? Does the sample statistically represent the population studied?
Response 1: Thank you for your valuable comment. In order to avoid self-plagiarism, the comprehensive description of the sample generation process has not been repeated here, but can be found in Kasdorf et al. 2024 (Dying at home: What is needed? Findings from a nationwide retrospective cross-sectional online survey of bereaved people in Germany). As not all federal states were represented to the desired extent, our sample cannot be considered statistically representative.
Comment 2: Could authors provide more information about the services provided to the deceased (and caregivers) during the care pathways? This comment is due to the different mix of aides provided by different services: for instance, home palliative care services could differ from geriatric home care services, especially regarding bureaucratic/administrative barriers, composition of the care teams and 24/7 assistance.
Response 2: Thank you very much for your insightful comment. You are absolutely right that there may be differences in the services provided. However, the common denominator for our study population is the expressed wish of the deceased to die at home. We did not carry out a comparative analysis of the different services offered, as this would have been beyond the scope of our current study.
Comment 3: Could authors provide any information about the dimensions of the trade-off regarding the resources provided to caregiver/potentially needed from the caregiver-patients?
Response 3: Thank you for your interesting question. It would certainly be interesting to explore quantitative data on the trade-offs between resources provided to caregivers and resources needed by both caregivers and patients. However, as this study focused primarily on the analysis of free text responses, we are not able to present this aspect.
Comment 4: I would suggest that authors should emphasize, in Strength and Limitations section, that since the survey has been conducted online, alphabetization on using online survey could affect differently according to the age. This aspect may explain partially the differences in age distribution of the sample, compared to other studies.
Response 4: Thank you for your suggestion. We agree and have added more information to this section.